# Multi-locus genotyping reveals established endemicity of a geographically distinct *Plasmodium vivax* population in Mauritania, West Africa

**Hampate Ba**[1☯]*, **Sarah Auburn**[2☯]*, **Christopher G. Jacob**[3], **Sonia Goncalves**[3], **Craig W. Duffy**[4], **Lindsay B. Stewart**[4], **Ric N. Price**[2], **Yacine Boubou Deh**[1], **Mamadou Yero Diallo**[1], **Abderahmane Tandia**[1], **Dominic P. Kwiatkowski**[3], **David J. Conway**[4]*

1 Institut National de Recherche en Santé Publique, Nouakchott, Mauritania, 2 Menzies School of Health Research and Charles Darwin University, Darwin, Australia, 3 Wellcome Sanger Institute, Hinxton, Cambridge, United Kingdom, 4 London School of Hygiene & Tropical Medicine, Keppel St, London, United Kingdom

☯ These authors contributed equally to this work.
* hampateba2001@yahoo.fr (HB); sarah.auburn@menzies.edu.au (SA); david.conway@lshtm.ac.uk (DJC)

**Data Availability Statement:** All relevant data are within the manuscript and its Supporting Information files.

## Abstract

### Background

*Plasmodium vivax* has been recently discovered as a significant cause of malaria in Mauritania, although very rare elsewhere in West Africa. It has not been known if this is a recently introduced or locally remnant parasite population, nor whether the genetic structure reflects epidemic or endemic transmission.

### Methodology/Principal findings

To investigate the *P. vivax* population genetic structure in Mauritania and compare with populations previously analysed elsewhere, multi-locus genotyping was undertaken on 100 clinical isolates, using a genome-wide panel of 38 single nucleotide polymorphisms (SNPs), plus seven SNPs in drug resistance genes. The Mauritanian *P. vivax* population is shown to be genetically diverse and divergent from populations elsewhere, indicated consistently by genetic distance matrix analysis, principal components analyses, and fixation indices. Only one isolate had a genotype clearly indicating recent importation, from a southeast Asian source. There was no linkage disequilibrium in the local parasite population, and only a small number of infections appeared to be closely genetically related, indicating that there is ongoing genetic recombination consistent with endemic transmission. The *P. vivax* diversity in a remote mining town was similar to that in the capital Nouakchott, with no indication of local substructure or of epidemic population structure. Drug resistance alleles were virtually absent in Mauritania, in contrast with *P. vivax* in other areas of the world.

### Conclusions/Significance

The molecular epidemiology indicates that there is long-standing endemic transmission that will be very challenging to eliminate. The virtual absence of drug resistance alleles suggests

**Funding:** This study was primarily funded by the UK Medical Research Council (MRC) Project Grant G1100123 to DJC and HB. Parasite genotyping was enabled by the SPOT-Malaria project through the MalariaGEN consortium led by DPK with funding from the Wellcome Trust. RNP is supported by the Wellcome Trust, SA is supported by the Bill and Melinda Gates Foundation, and a Georgina Sweet Award for Women in Quantitative Biomedical Science. The funders had no role in study design, data collection and analysis, decision to publish, or preparation of the manuscript.

**Competing interests:** The authors have declared that no competing interests exist.

that most infections have been untreated, and that this endemic infection has been more neglected in comparison to *P. vivax* elsewhere.

## Author summary

*Plasmodium vivax* is a widespread cause of malaria in Mauritania, in contrast to its rarity elsewhere throughout West Africa. To investigate whether the parasite may be recently introduced or epidemic, multi-locus genotyping was performed on 100 Mauritanian *P. vivax* malaria cases. Analysis of a genome-wide panel of single nucleotide polymorphisms showed the *P. vivax* population to be genetically diverse and divergent from populations elsewhere, indicating that there has been long-standing endemic transmission. Almost all infections appear to be locally acquired, with the exception of one that was presumably imported with a genotype similar to infections seen in Southeast Asia. The Mauritanian *P. vivax* population shows no linkage disequilibrium, and very few infections have closely related genotypes, indicating ongoing recombination. The parasite showed no indication of local substructure or epidemic population structure. Drug resistance alleles were virtually absent, suggesting that most infections have been untreated historically. The molecular epidemiology indicates that there has been long-standing endemic transmission of this neglected parasite that requires special attention for control.

## Introduction

On the edge of the Sahara in northwest Africa, Mauritania is endemic for malaria, with the majority of cases normally attributed to *Plasmodium falciparum* [1]. Analysis of parasite DNA from clinical samples indicates that most malaria in the Sahel zone in the south of the country is caused by *P. falciparum* [2], whereas *P. vivax* predominates in the central and northern regions of the country within the arid Saharan zone [2–4]. Routine laboratory-based diagnostic testing is conducted in health centres where facilities allow, using either slide microscopy or rapid diagnostic tests, although presumptive clinical diagnosis has also been commonly applied. Chloroquine was first-line treatment for uncomplicated clinical malaria in Mauritania until 2006, with sulphadoxine-pyrimethamine being only occasionally used, and since then artemisinin combination therapy with artesunate-amodiaquine has been officially first-line treatment for all uncomplicated malaria cases, while sulphadoxine-pyrimethamine is reserved for intermittent preventive treatment during pregnancy [3].

There is evidence suggesting that *P. vivax* in humans emerged in Africa as a spill-over zoonosis from a reservoir in great apes [5,6], but this parasite species is now rare in most of the continent apart from the Horn of East Africa [7,8] and some analyses suggest historical emergence from southeast Asia [9,10], so the origin and epidemiology in northwest Africa remains obscure. It is vital to know whether *P. vivax* has emerged in Mauritania in the recent past, as was assumed in the initial report of its presence in the country [11], or whether this is a long-established endemic parasite whose presence has only been recognised recently.

Genotypic characterization can provide important information on the origins of parasite populations and their relationships to those elsewhere, and can also illuminate local population structure and variation in transmission [12], as well as the prevalence and spread of drug resistance [13]. Here, we employed multi-locus single nucleotide polymorphism (SNP) analysis of a genome-wide set of loci [14] to analyse *P. vivax* population genetic structure in patient samples from three areas in Mauritania where this species was noted to be the most common

cause of malaria [2], and also analysed candidate drug resistance markers in the same samples. In comparison with genotypic data on *P. vivax* populations in other countries, the *P. vivax* population in Mauritania is distinct, and almost all of the infections appeared to have been acquired within the country. The *P. vivax* population in Mauritania showed no evidence of genetic sub-structure, and is apparently highly recombining, consistent with long-established and ongoing endemicity. Controlling this infection requires characterization of the epidemiology in areas that have had minimal attention previously, and a commitment to radical cure will be important for regional elimination.

## Methods

### Ethics Statement

All samples were obtained with written informed consent from patients with malaria presenting for treatment, as well as guardians of any patients who were under 18 years of age. The study was approved by ethics committees of the Ministry of Health of Mauritania and the London School of Hygiene and Tropical Medicine (Ethics approval number 6043).

### *Plasmodium vivax* clinical infection samples

*P. vivax* infections from patients presenting to local health centres and hospitals in different parts of Mauritania in 2012 and 2013 were previously identified by slide microscopy [15] and confirmed by species-specific PCR of genomic DNA from samples collected on filter paper [2]. All cases were local residents who did not report having travelled within the previous two weeks, and who were invited to provide finger-prick blood samples which were collected on filter paper and air-dried prior to storage with desiccant in sealed polythene bags. All samples were obtained with informed consent from patients, and guardians of patients who were under 18 years of age. After molecular confirmation of species, DNA from 100 isolates was analysed successfully by multi-locus SNP genotyping. Most of these (N = 91) were from the capital city Nouakchott, located on the Atlantic coast in the southern edge of the Sahara zone with low annual rainfall (100–200 mm), and dependent on a supply of piped water from the Senegal River basin in the south of the country. A smaller number of genotyped samples (N = 8) were from Zouérat, an iron ore mining town in the extremely arid northern Sahara zone of the country (receiving less than 50 mm annual rainfall), all cases being local residents as the mining operation is long-established and supports a developed urban area with a settled economy and extended families. A single genotyped sample was from N'beika, a small oasis town in the Sahara zone in the central part of the country (receiving 50–200 mm annual rainfall) (Fig 1). All of the *P. vivax* cases in this study were from individuals with Duffy positive genotypes as determined by PCR analysis previously [2], reflecting that this parasite species is mainly seen in areas of the country where most of the population are of the majority Maure ethnicity.

### Multi-locus SNP genotyping

Genotyping was undertaken for a panel of 38 SNP loci from across the *P. vivax* genome (including markers on all of the 14 chromosomes), representing 90% of the SNPs on a previously described array of putatively neutral markers [14] for which amplicon-based sequencing assays were used in genotyping here (S1 Table). In addition, seven SNPs that have been associated with drug resistance elsewhere were genotyped, in the genes encoding dihydropteroate synthase (*dhps*, codon polymorphisms A553G and A383G), dihydrofolate reductase (*dhfr*, codon polymorphisms F57L/I, S58R, T61M and S117N/T), and the multidrug resistance 1 locus (*mdr1*, codon Y976F) [16]. Genotyping was performed using amplicon-based

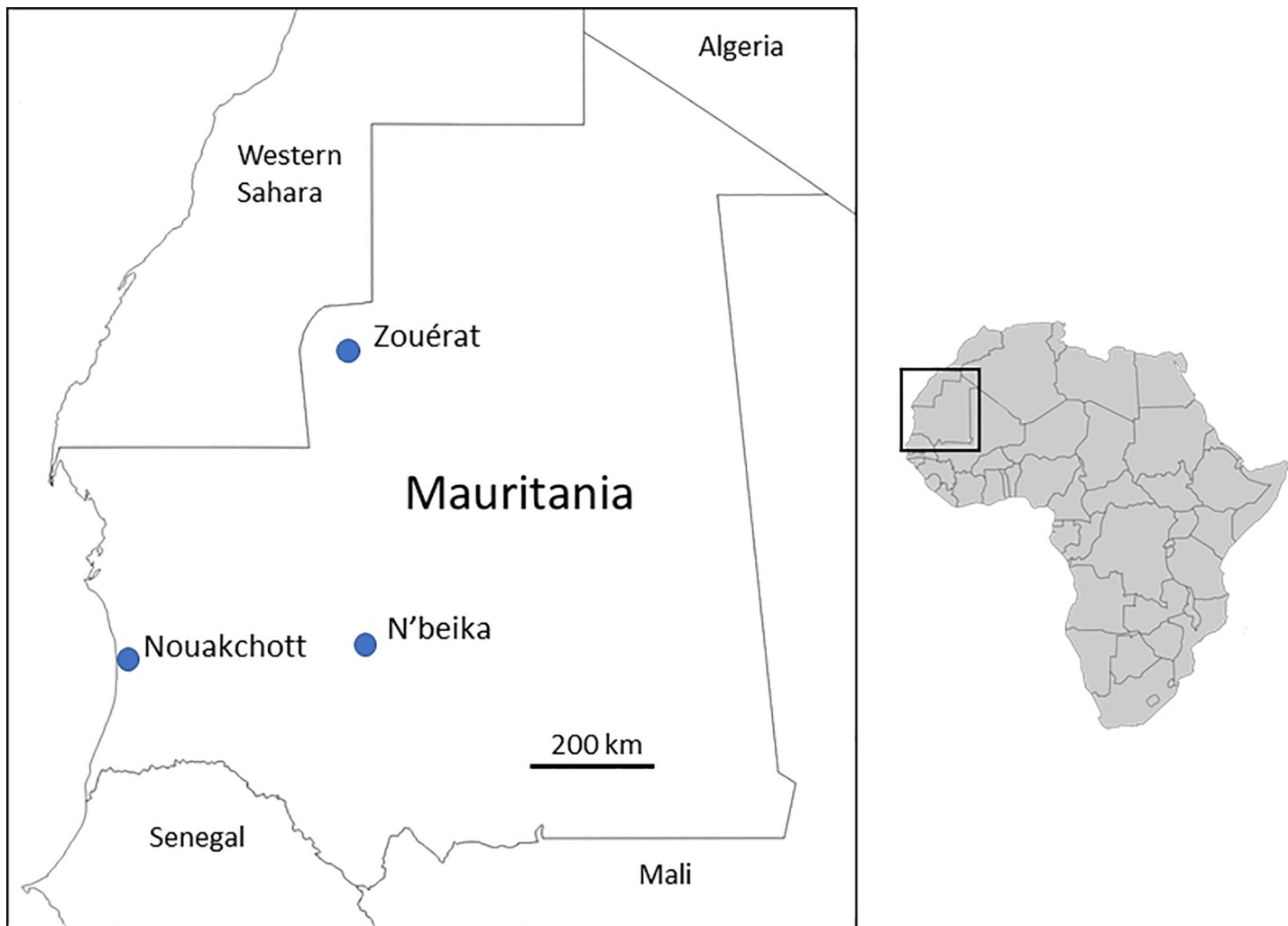

**Fig 1. Map showing the location of the sampling sites of 100 *P. vivax* malaria cases in Mauritania analysed by multi-locus genotyping in this study.** Most cases analysed were from the capital city (Nouakchott, N = 91), while eight were from a northern mining town (Zouérat), and one was from a central oasis town (N'beika). The map of Africa to the right shows the location of Mauritania and surrounding countries in northwest Africa illustrated in the main map.

sequencing on the Illumina MiSeq platform at the Wellcome Sanger Institute, with reference alleles based on the PvP01 *P. vivax* reference genome sequence [17]. Individual infection samples were considered valid for analysis if clear genotyping scores could be obtained for more than half of the SNP loci, yielding a total of 100 infections for analysis (in most of these samples complete data were obtained for all SNP loci as noted in the Results).

To compare *P. vivax* from Mauritania with parasite populations in other countries, the SNP genotypes for these same loci were derived using VCFtools from previously published genome sequence data from *P. vivax* clinical isolates collected in Ethiopia (2013, N = 24), Thailand (2006–2013, N = 104), Papua Province in Indonesia (2011–2014, n = 111) [18–20], Mexico (2000–2007, n = 20) [21], and Colombia (2012–2013, n = 31) [21].

## Population genetic and statistical analysis

The population genetic structure analyses focused on the 38 putatively neutral SNPs from throughout the *P. vivax* genome, while the seven drug resistance SNPs were analysed

separately. If mixed alleles at a particular SNP were detected in an infection sample, the major one (with the highest read depth) was scored to contribute to the multi-locus genotype profile, and if this was not clear the SNP call was recorded as missing. A pairwise measure of genetic distance between infections (1-$ps$) considered the proportion of alleles shared ($ps$), and a Neighbor-Joining tree of the distance matrix was generated using the ape package in R and iTOL software [22,23]. Similarity of individual samples was also assessed using Principal Component Analysis (PCA) using the ClustVis software (https://biit.cs.ut.ee/clustvis). The degree of divergence between Mauritania and other geographical populations was assessed using the fixation index ($F_{ST}$), employing the Scikitallel package (https://github.com/cggh/scikit-allel) to determine Hudson's $F_{ST}$ estimator, and using an in-house R script to calculate Weir and Cockerham's $F_{ST}$ as these different estimators focus on slightly different components of the allele frequency spectra [24]. Within-population SNP diversity was assessed using the virtual heterozygosity index ($H_e$), the mean of the pairwise differences at each marker between isolates within a given population. A within-population multi-locus measure of statistical linkage disequilibrium (LD) among the SNP loci was provided by the Index of Association ($I^S_A$), calculated for samples with complete multi-locus genotype data using LIAN version 3.7 software [25]. This measure of LD was assessed by analysing all samples that included mixed infections, as well as only in samples that had single genotype infections, and with unique multi-locus genotypes. The single genotype infections were defined as those that yielded a Complexity of Infection of 1 using the COIL algorithm [26]. R software was used for statistical tests, including Pearson's Chi-squared test with Yates' continuity correction for analysis of categorical variables.

## Results

### High-quality multi-locus SNP genotyping of Mauritanian *P. vivax* samples

Of the 100 Mauritanian *P. vivax* infection samples that had sufficient DNA for genotyping in this study, 84 yielded complete SNP data for all loci (38 loci genome-wide plus seven SNPs in drug resistance genes), and only six samples had >30% SNPs drop out. Twelve of the 100 infection samples contained multiple genotypes at any of the SNP loci, hence the haploid multi-locus genotype could be resolved unequivocally for 88 infections, of which 80 had complete data for all SNPs (all multi-locus genotypes for all individual samples are shown in S1 Datasheet).

### Population genetic diversity of *P. vivax* in Mauritania and divergence from other populations

Based on the array of 38 putatively neutral SNP markers from throughout the genome, there was a high level of diversity in Mauritania, assessed by the virtual heterozygosity index of pairwise differences among all infections averaged across all SNPs ($H_e$ = 0.31). Analysis of the pairwise genetic differences revealed that almost all infections were different from each other. The samples from Zouérat and N'beika were distributed in the same overall range of diversity amongst those from Nouakchott, indicating no genetically separated parasite subpopulations within the country. With exception of three infections from Nouakchott, the Mauritanian samples formed a distinct geographical group, with slightly closer relatedness to previous samples from Ethiopia as well as Central and South America than to most samples from Southeast Asia (Fig 2 and S1 Fig). Of the three outlier infections, two were only slightly outside the grouping of most Mauritanian samples, but one (isolate NKT136 from Nouakchott) was closely related to parasites previously sampled from Southeast Asia.

Among all of the Mauritanian samples, only three pairs of infections had identical multi-locus genotypes (Fig 2). These were not epidemiologically linked, NKT144 and TY004 having

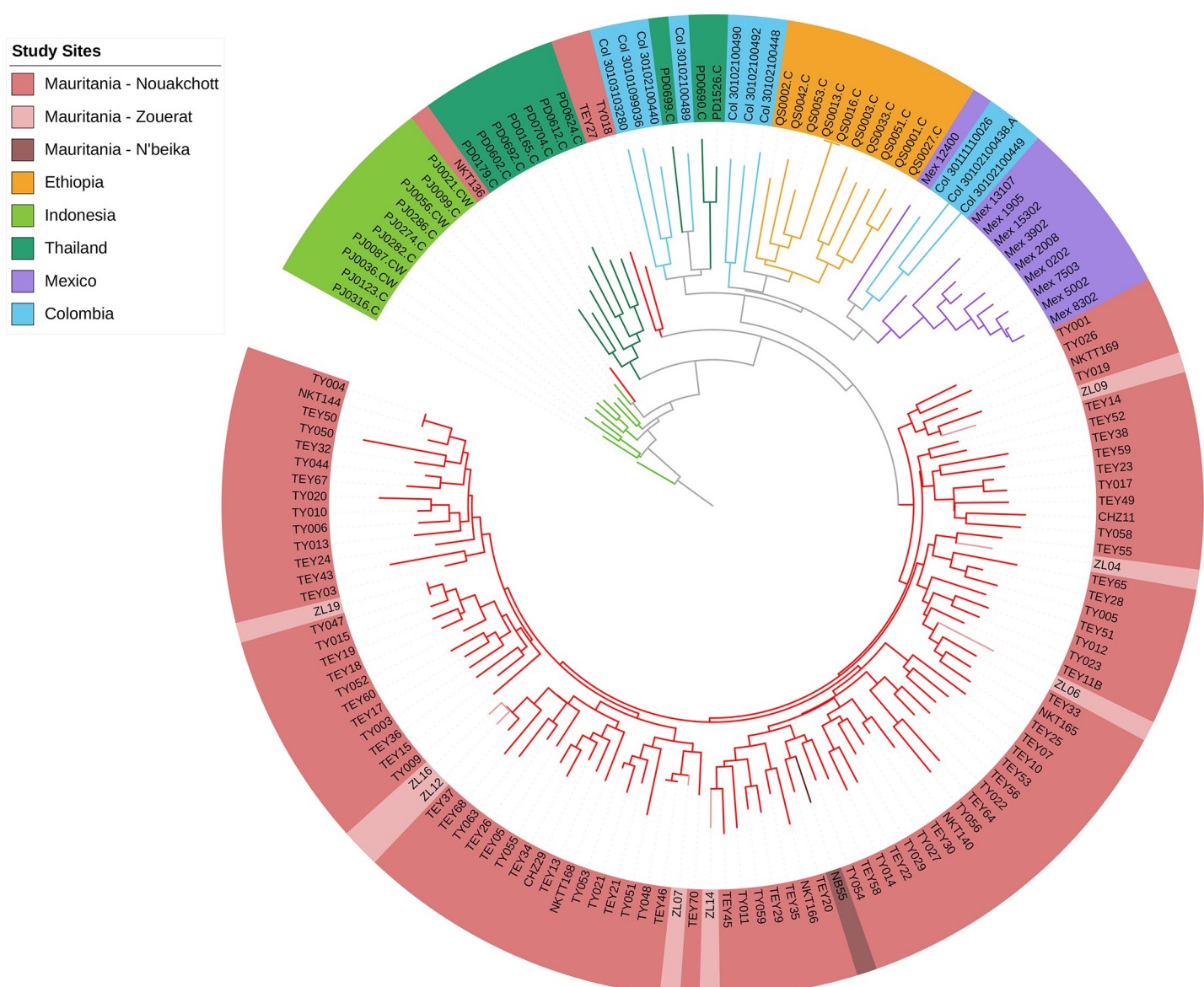

**Fig 2. Neighbor-Joining tree illustrating the genetic relatedness among *P. vivax* infections in Mauritania in comparison with previous samples from other countries, based on multi-locus genotypes based on a distance matrix using the genome-wide panel of 38 SNPs genotyped in this study.** Data are presented on all samples from Mauritania (n = 100) and a random selection of 10 samples from each of Ethiopia, Thailand, Indonesia, Mexico and Colombia. For visual clarity, a rooted tree is shown using an arbitrarily selected isolate from Indonesia (PJ0316-C) as the 'root' (note that this is not a phylogenetic tree but a genetic distance dendrogram of a recombining species, and an unrooted tree has similar topology shown in S1 Fig). Within Mauritania, isolates from the mining town Zouérat and the oasis town N'beika were not genetically separated from samples from the capital city Nouakchott. Mauritanian isolates are diverse but comprise a population distinct from those in the other countries, except for two isolates that are similarly divergent as those from Ethiopia and one isolate (NKT136), showing closer genetic relatedness to the Asian parasites.

been sampled 123 days apart in Nouakchott, TEY46 and TY048 sampled 312 days apart in Nouakchott, and TY047 and ZL019 respectively sampled 210 days apart in Nouakchott and Zouérat. There was no significant multi-locus linkage disequilibrium in the total population sample from Mauritania ($I_A^S$ = 0.002, $P$ > 0.05), or separately within Nouakchott ($I_A^S$ = 0.002, $P$ > 0.05) or Zouérat ($I_A^S$ = -0.012, $P$ > 0.05). Multi-locus LD remained low in Mauritania when analysis was restricted to single genotype infections ($I_A^S$ = 0.0004, $P$ > 0.05) and infections with unique multi-locus genotypes ($I_A^S$ = 0.002, $P$ > 0.05). This is consistent with

established endemicity and ongoing genetic recombination in the local *P. vivax* population, without any evidence of clonal or epidemic population structure which is seen elsewhere when local transmission is unstable (S2 Table).

Principal Component Analysis confirmed that almost all of the Mauritanian *P. vivax* infection samples clustered separately from those previously sampled in other countries, although they were more closely related to those from Ethiopia than from other continents (Fig 3). Consistent with the Neighbor-Joining analysis, it showed that one infection sample (NKT136 from Nouakchott) was genetically similar to parasites from Asia. Further Principal Component Analyses comparing Mauritania with each of the other countries separately, confirmed that all *P. vivax* samples from Mauritania clustered separately from all others (except isolate NKT136 which consistently clustered with Asian *P. vivax* samples) (S2 Fig).

The genetic differentiation between the Mauritanian *P. vivax* population and those from other countries was also estimated using the inter-population fixation index $F_{ST}$ based on allele frequencies, revealing a high level of divergence from Ethiopia ($F_{ST} = 0.22$ using Hudson's

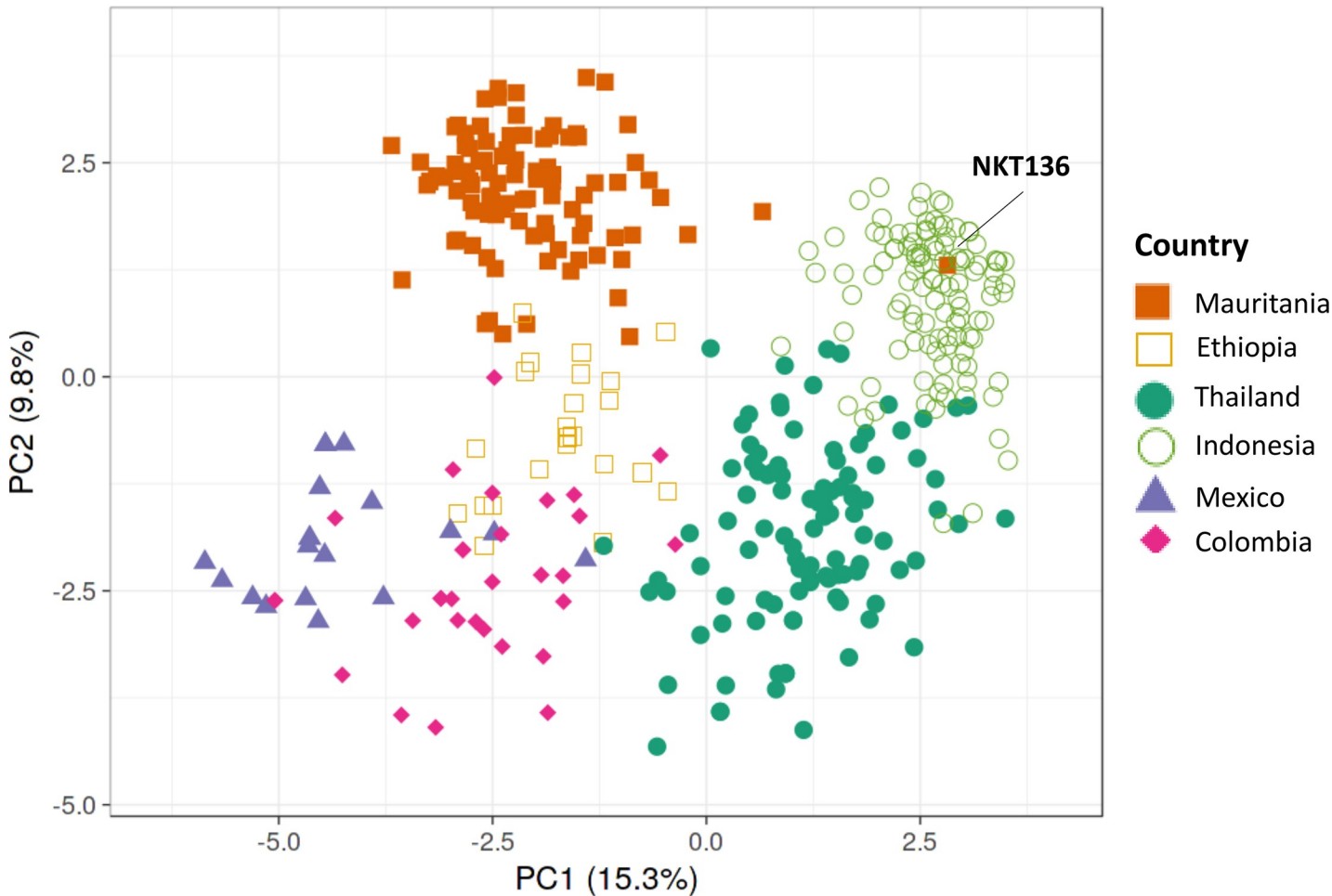

**Fig 3. Principal Component Analysis (PCA) plot illustrating genetic relatedness among *P. vivax* infections in Mauritania compared with previous data from other endemic countries [18–20].** The first two principal components are shown, illustrating that the parasite population in Mauritania is distinct from the Southeast Asian and Central and South American populations, and has only a minor degree of overlap with samples from Ethiopia. In agreement with Neighbour-Joining analysis of genetic distance (shown in Fig 2), the single isolate NKT136 from Nouakchott clustered with parasites from Southeast Asia. Further Principal Component Analysis of the Mauritanian data with each of the other populations separately confirm the population distinctness (with the exception of isolate NKT136), and also show that isolates from different sampled areas of Mauritania are not genetically separated (S2 Fig).

estimator) and even higher divergence from populations previously sampled in other continents (S3 Table, also shows $F_{ST}$ values using Weir and Cockerham's estimator with slightly lower values but the same trend).

### Extremely low frequency of drug resistance alleles in *P. vivax* in Mauritania

The proportion of infections containing drug resistance-associated SNP alleles was investigated, and shown to be exceptionally low compared with data from other geographical populations of *P. vivax* (Table 1). Notably, there were no antifolate resistance-associated alleles at codons 57, 58 and 61 of the dihydrofolate reductase (*dhfr*) gene in Mauritania (Table 1). Alleles at codon 58 were common in most other *P. vivax* populations including Ethiopia ($P < 1$ x $10^{-15}$ for each comparison with Mauritania), and at codons 57 and 61 were very common in the Asian populations ($P < 1$ x $10^{-15}$ for each comparison with Mauritania). One infection in Mauritania had a resistance-associated allele at codon 117 of *dhfr* (S117N), but this was infection sample NKT136 that was shown by the multi-locus analysis of neutral SNPs to be genetically similar to Asian parasites, so the occurrence of this allele is likely to reflect an exceptional case of imported malaria (individuals who had travelled within the two weeks prior to diagnosis were not included in the study but earlier travel was not recorded).

At the dihydropteroate synthase (*dhps*) gene, there were no resistance-associated alleles in Mauritania at codon 553 (Table 1), and a resistance-associated allele at codon 383 was only detected in two infections, one of which was NKT136 that has a genotypic profile of Asian *P. vivax* as noted above. The virtual absence of resistance-associated *dhps* alleles in Mauritania contrasts with other populations, including Ethiopia where such alleles are of moderately low frequency ($P = 0.021$), and populations in other continents where they are very common ($P < 1$ x $10^{-5}$ for each comparison, except for codon 383 in Mexico) (Table 1). In the multi-drug resistance gene (*mdr1*), the Y976F variant which has been loosely associated with chloroquine resistance elsewhere [27] was detected in 14% of infections in Mauritania (Table 1).

## Discussion

Although *P. vivax* has only been recognised as a significant cause of malaria in Mauritania recently, this study indicates that the parasite has been long-established in northwest Africa, and that it is a genetically distinctive sub-population compared with parasites from other areas of the world. A genome-wide panel of 38 SNP loci revealed a high level of *P. vivax* genetic diversity in Mauritania, and no evidence of linkage disequilibrium or population genetic substructure within the country. The *P. vivax* population in Mauritania is clearly genetically

**Table 1. Low frequency of SNP variants in *P. vivax* drug resistance genes in Mauritania compared with other endemic populations.**

| Gene | Codon Allele | Percentage of isolates containing resistance-associated variants (with numbers in brackets) in: | | | | | |
|------|--------------|------------|------------|------------|------------|------------|------------|
| | | Mauritania | Ethiopia | Thailand | Indonesia | Mexico | Colombia |
| *dhfr* | F57L/I | 0 (0/89) | 0 (0/24) | 91 (88/97) | 82 (77/94) | 0 (0/20) | 0 (0/31) |
| | S58R | 0 (0/87) | 94 (17/18) | 100 (104/104) | 99 (104/105) | 0 (0/20) | 97 (30/31) |
| | T61M | 0 (0/90) | 0 (0/24) | 91 (90/99) | 82 (77/94) | 0 (0/20) | 0 (0/31) |
| | S117N/T | 1 (1/98) | 100 (24/24) | 100 (102/102) | 99 (93/94) | 0 (0/20) | 97 (30/31) |
| *dhps* | A553G | 0 (0/99) | 0 (0/24) | 98 (98/100) | 16 (16/97) | 100 (20/20) | 100 (31/31) |
| | A383G | 2 (2/100) | 17 (4/24) | 100 (104/104) | 97 (106/109) | 0 (0/20) | 84 (26/31) |
| *mdr1* | Y976F | 14 (14/100) | 32 (6/19) | 13 (14/104) | 100 (111/111) | 0 (0/19) | 3 (1/29) |

Data for Mauritania are from the current study, compared alongside previously published data from Ethiopia [19], Thailand [18,20], Indonesia [18], Mexico [21] and Colombia [21].

divergent from other geographical populations of the parasite, including Ethiopia which is the source of most samples from the African continent that have been analysed by sequencing [19,28,29]. A recent survey of 14 microsatellite loci has also indicated the distinctiveness of *P. vivax* samples from Mauritania compared to most other sampled populations in Africa, Asia or the Americas [10], further supporting the evidence for an established focus of endemicity.

It is notable that there is no evidence for a clonal or epidemic population genetic structure of *P. vivax* in Mauritania, whereas such structure has been seen in populations close to elimination, such as described in Malaysia [30] and parts of Central and South America [21]. The areas where *P. vivax* infections were sampled in Mauritania experience extremely low annual rainfall, within only a few months of the year at most and sometimes none, and this would clearly limit opportunities for transmission by mosquitoes. The other endemic species *P. falciparum* has been shown to occasionally have an epidemic population structure at a minority of sites sampled further south in the country where rainfall is not quite so low [31]. As *P. vivax* showed no evidence of a clonal or epidemic structure in extremely dry areas studied here, it is likely that many infections persist for multiple years due to latency of liver-stage parasites, occasional relapses allowing ongoing transmission to occur when conditions allow. For future control and to aim towards eventual elimination, radical cure of *P. vivax* infections using either primaquine or tafenoquine will be needed, which would require screening for glucose 6-phosphate dehydrogenase deficiency of patients to reduce the risk of haemolytic anaemia as a side effect [32].

This northwest African *P. vivax* population has probably been present since pre-history, and may have previously been part of a wider geographical population. To the north, *P. vivax* was eliminated from Morocco in 2010, and used to be present in neighbouring Algeria where it is now almost eliminated [1,33]. This may even be part of the same parasite population that was endemic in Europe until malaria was eliminated in the mid-twentieth century. Sequencing of archival *P. vivax* from Europe could be performed, as illustrated by analysis of a single specimen so far [34], and once there are data from a range of other samples their relatedness to sequences from parasites in Mauritania should be determined. It is an ongoing priority to increase the amount of *P. vivax* whole-genome sequence data from infections in Mauritania and other countries where the epidemiology and population history is unclear [35].

To the east and south of Mauritania, a small number of *P. vivax* infections have been documented in Mali and Senegal respectively [36–38], and it is now considered that this species may occur at low densities more widely in Africa than previously assumed [39,40]. It will be important to genotype *P. vivax* from other African countries to ascertain if there are any other endemic populations, and any connectivity among them, or if infections elsewhere reflect sporadic introductions. The substantial genetic divergence we show between *P. vivax* in Mauritania and Ethiopia indicates that there is little gene flow between these two main endemic foci in Africa, consistent with the parasite having only a very sparse and discontinuous endemic distribution elsewhere on the continent.

Analysis of *dhfr* and *dhps* genotypes revealed a virtual absence of antifolate resistance-associated variants in *P. vivax* in Mauritania. Consistent with this, drug resistance genotyping of *P. vivax* samples previously taken from Nouakchott in 2007–9 and 2013–16 also showed no resistance-associated alleles in *dhps* [41,42]. However, the same studies reported between 10 and 20% of infections to have double mutant alleles of *dhfr* (combinations of variants at codons 58, 61 and 117) [41,42], in contrast to results here that showed an absence of resistance-associated alleles at these codons (except for a single variant at codon 117 in infection sample NKT136 that had an overall genotype similar to Asian parasites). Antimalarial antifolate use in Mauritania is now limited to intermittent preventative treatment of pregnant women with sulphadoxine-pyrimethamine [41], although antifolates were previously used for therapy along with

chloroquine. Comparing this and previous studies, the frequency of the *mdr1* 976F allele in Mauritania varied from 28% in 2007–9 to 14% in 2012–13, and 4% in 2013–16 [41,42], which may reflect a decline due to fitness cost after chloroquine use officially stopped following introduction of Artemisinin Combination Therapy in 2006, although the allele is not a marker of chloroquine resistance and is at most a minor modulator [16]. Clinical evidence indicates that most local *P. vivax* infections are sensitive to chloroquine treatment [43].

In line with political and public health aspirations, Mauritania is aiming to eliminate malaria by 2030, but for this to be attainable a much deeper understanding of *P. vivax* in the region will be essential, as this species is particularly difficult to eliminate [1,44,45]. The distinct genetic profile of Mauritanian *P. vivax* compared with parasites elsewhere has important implications. It is highly likely that this population has locally adapted, to maintain parasites in the extreme conditions of the Sahara where opportunities for mosquito vector transmission must be exceedingly rare. Further epidemiological monitoring, clinical studies and genomic analyses of parasites from this unusual population will be necessary to understand this highly important parasite species that has been so long overlooked in Africa.

## Supporting information

**S1 Fig. Unrooted Neighbor-Joining tree illustrating the genetic relatedness among *P. vivax* infections in Mauritania and previous samples from other countries, as described in the Methods and Results, based on a distance matrix of multi-locus genotypes using the panel of 38 SNPs genotyped in this study.** Note that the topology is similar to the rooted tree in Fig 2, and a single isolate from Mauritania (NKT136) has a genotype similar to Southeast Asian parasites. Note that this is not a phylogenetic tree but a genetic distance dendrogram of a recombining species.
(TIFF)

**S2 Fig.** PCA of *P. vivax* genotypes (38-SNP array) of individual clinical isolates from different pairs of sites, comparing data from Mauritania in this study with data previously published from elsewhere: (a) Within Mauritania (Zouerat, N'beika and Nouakchott), (b) Mauritania and Ethiopia, (c) Mauritania and Thailand, (d) Mauritania and Indonesia, e) Mauritania and Mexico, f) Mauritania and Colombia. This shows no genetic separation of parasites from different sites within Mauritania, but separation of the Mauritanian population from parasites in each of the other countries (previous data from Ethiopia, Thailand, Indonesia, Mexico and Colombia as cited in the Methods and Results). The outlier isolate NKT136 which shows genetic similarity to those from Southeast Asia is circled.
(TIFF)

**S1 Table. List of 38 *P. vivax* SNP marker loci analysed in this study.**
(DOCX)

**S2 Table. Multi-locus linkage disequilibrium in *P. vivax* in Mauritania and in previous data from other countries.**
(DOCX)

**S3 Table. Genetic differentiation between the Mauritanian *P. vivax* population sampled in this study and previously described populations.**
(DOCX)

**S1 Datasheet. Genotypes of each of the 100 Mauritanian isolates in this study for the array of 38 SNP loci.**
(XLS)

## Acknowledgments

We are grateful to all patients, and staff of the health facilities for willing participation and support throughout the surveys. We acknowledge the support and encouragement of the Director and other colleagues at the Institut National de Recherche en Santé Publique in Mauritania, and we also thank colleagues at the London School of Hygiene and Tropical Medicine and the Wellcome Sanger Institute for support. We also thank Hidayat Trimarsanto for assistance in compiling published genomic datasets for comparisons.

## Author Contributions

**Conceptualization:** Hampate Ba, David J. Conway.

**Data curation:** Hampate Ba, Sarah Auburn, Christopher G. Jacob, Sonia Goncalves, Dominic P. Kwiatkowski, David J. Conway.

**Formal analysis:** Sarah Auburn, David J. Conway.

**Funding acquisition:** Hampate Ba, Sarah Auburn, Ric N. Price, Dominic P. Kwiatkowski, David J. Conway.

**Investigation:** Hampate Ba, Sarah Auburn, Christopher G. Jacob, Sonia Goncalves, Craig W. Duffy, Lindsay B. Stewart, Yacine Boubou Deh, Mamadou Yero Diallo, Abderahmane Tandia, David J. Conway.

**Methodology:** Sarah Auburn, Christopher G. Jacob, Sonia Goncalves, Dominic P. Kwiatkowski, David J. Conway.

**Project administration:** Hampate Ba, David J. Conway.

**Resources:** Dominic P. Kwiatkowski, David J. Conway.

**Supervision:** Hampate Ba, Dominic P. Kwiatkowski, David J. Conway.

**Validation:** Sarah Auburn, Christopher G. Jacob, Sonia Goncalves, Dominic P. Kwiatkowski, David J. Conway.

**Writing – original draft:** Hampate Ba, Sarah Auburn, David J. Conway.

**Writing – review & editing:** Hampate Ba, Sarah Auburn, Christopher G. Jacob, Sonia Goncalves, Ric N. Price, Dominic P. Kwiatkowski, David J. Conway.

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
