## [Decision Letter · Decision Letter 0]

15 Oct 2020

Dear Prof. Conway,

Thank you very much for submitting your manuscript "Multi-locus genotyping reveals established endemicity of a geographically distinct Plasmodium vivax population in Mauritania, West Africa" for consideration at PLOS Neglected Tropical Diseases. As with all papers reviewed by the journal, your manuscript was reviewed by members of the editorial board and by several independent reviewers. The reviewers appreciated the attention to an important topic. Based on the reviews, we are likely to accept this manuscript for publication, providing that you modify the manuscript according to the review recommendations. 

Sincerely,

Adalgisa Caccone

Associate Editor

Mary Lopez-Perez

Deputy Editor

Reviewer's Responses to Questions

**Key Review Criteria Required for Acceptance?**

**Methods**

-Are the objectives of the study clearly articulated with a clear testable hypothesis stated?

-Is the study design appropriate to address the stated objectives?

-Is the population clearly described and appropriate for the hypothesis being tested?

-Is the sample size sufficient to ensure adequate power to address the hypothesis being tested?

-Were correct statistical analysis used to support conclusions?

-Are there concerns about ethical or regulatory requirements being met?

Reviewer #1: 186/5000

Include in the Introduction the treatment scheme for Plasmodium vivax in the region and information to support the drug resistance-associated SNP alleles studied. The objectives are well outlined in the face of the problem presented. The methodology described is adequate to achieve the proposed objectives. The sample allows for the proposed investigation. Statistical analyzes are adequate. All samples were obtained with written informed consent from patients with malaria

162 presenting for treatment, as well as guardians of any patients who were under 18 years of

163 acts. The study was approved by ethics committees of the Ministry of Health of Mauritania

164 and the London School of Hygiene and Tropical Medicine (Ethics approval number 6043).

Reviewer #2: The objectives of the study are clearly articulated. The study design is appropriate to the objectives. A few additional details would be welcome:

1. What is the primary method of malaria diagnosis performed in Mauritania (microscopy or RDTs)? If RDTs, is the standard a Plasmodium falciparum specific RDT?

2. It would be recommended to address the human genetic differences in Mauritania (similar to Ethiopia) that partially contribute to the high prevalence relative to other regions in sub-Saharan Africa and also differentiate between infection in Duffy positive individuals in Mauritania in contrast with the accounts of Duffy negative P. vivax infection in other SSA countries.

**Results**

-Does the analysis presented match the analysis plan?

-Are the results clearly and completely presented?

-Are the figures (Tables, Images) of sufficient quality for clarity?

Reviewer #1: The results are well presented with very informative tables. Improve the map (Fig 1) of Mauritania to locate the area on the respective continente.

Reviewer #2: The results are clearly and completely presented and the analysis presented is appropriate given the analysis plan. 

It would be interesting to have more details in the following areas:

• While the individuals in the study reported no travel, more detail on the samples from Zouérat, the mining town would be welcome. Did these samples come from residents or migratory miners? The results indicate that these populations were interspersed with the samples from Nouakchott in terms of genetic diversity, but it would be interesting to have some more details of the underlying patient population from this unique site.

• Lines 225-227: Both genetic analysis and drug resistance allele indicate a single sample of imported malaria from Asia. Would it be possible to epidemiologically track the origins of this importation or at least to provide more patient metadata that might clarify when this parasite entered Mauritania and how?

• The authors note that the samples were genetically diverse and only three pairs of infections had identical multi locus genotypes and were not epidemiologically linked. The authors comment on the temporal spacing of the samples with identical multi-locus genotypes, but has fine spatial mapping been performed to determine if identical samples in Nouakchott might show any neighborhood or household clustering patterns?

**Conclusions**

-Are the conclusions supported by the data presented?

-Are the limitations of analysis clearly described?

-Do the authors discuss how these data can be helpful to advance our understanding of the topic under study?

-Is public health relevance addressed?

Reviewer #1: Você quis dizer: As conclusões suportam de forma adequada os resultados obtidos

63/5000

The conclusions adequately support the results obtained. I suggest the authors to correlate the data found based on the data observed in the manuscrip “The Evolutionary History of Plasmodium vivax as Inferred from Mitochondrial Genomes: Parasite Genetic Diversity in the Americas. Taylor et al . Mol Biol Evol. 2013 Sep; 30 (9): 2050–2064”.

Reviewer #2: The conclusions are supported by the data. 

• It would be interesting if the authors could provide some clinical context on what is the standard procedure for detection and treatment of P. vivax in Mauritania and to comment of hypothesize why it has gone so largely undetected and untreated relative to other global populations.

• It would be helpful to have a little more data/clarification for the conclusion that "endemic infection (P. vivax) has been more neglected in comparison to P. falciparum locally”. Have comparisons by this group or others been specifically performed between Pf and Pv in this population?

• Lines 302-310: The authors speculate that given the rare rainfall that it is likely that most of the P. vivax infections are from relapses. Given this, what would be the most appropriate strategy for Mauritania to achieve malaria elimination and what would the caveats be with this approach? The paper has clearly and convincingly described the history and nature of population structure in Mauritania; however, a more detailed discussion of how these data can be helpful to advance our understanding and resulting intervention would be helpful for the public health relevance.

**Editorial and Data Presentation Modifications?**

Reviewer #1: Overall, the manuscript presents relevant information for those working with epidemiology, population genetics and malaria treatment by Plasmodium vivax. The study aims to investigate the P. vivax population genetic structure in Mauritania and compare with populations previously analyzed elsewhere, multi-locus genotyping. The work is able to be published in PLOS Neglected Tropical Diseases after minor revisions

Reviewer #2: The only potential modifications might be additional detail on the patient populations and the related epidemiology.

**Summary and General Comments**

Reviewer #1: Plasmodium vivax is responsible for most morbidity outside Africa. This parasite has reemerged in many regions of the world where malaria was eliminated in the 1950–60s. However, ; new evidence suggests that severe complications from P. vivax malaria may be more common than previously thought . Several lines of evidence are consistent with growth or expansion of P. vivax populations in most regions. The work is robust, well presented and provides relevant information for the study of malaria by this protozoan in the studied area, as well as for other endemic regions around the world.

Reviewer #2: Here, Ba et al combine epidemiology, genomics, and population genetics to understand two key questions about P. vivax malaria in Mauritania: Whether the P. vivax population is recently introduced or locally remnant and whether the genetic structure reflects epidemic or endemic transmission? The study is well designed, the analyses are clear and comprehensive, and the conclusions are supported by the data. Overall, this is a well designed and executed study that adds significant knowledge to our understanding of a unique niche of Plasmodium vivax in Sub-Saharan Africa.

PLOS authors have the option to publish the peer review history of their article (what does this mean?). If published, this will include your full peer review and any attached files.

Reviewer #1: No

Reviewer #2: No
---

## [Editor Report · Decision Letter 1]

3 Nov 2020

Dear Prof. Conway,

We are pleased to inform you that your manuscript 'Multi-locus genotyping reveals established endemicity of a geographically distinct Plasmodium vivax population in Mauritania, West Africa' has been provisionally accepted for publication in PLOS Neglected Tropical Diseases.

Best regards,

Adalgisa Caccone

Associate Editor

Mary Lopez-Perez

Deputy Editor

---

## [Editor Report · Acceptance letter]

28 Nov 2020

Dear Prof. Conway,

We are delighted to inform you that your manuscript, "Multi-locus genotyping reveals established endemicity of a geographically distinct *Plasmodium vivax* population in Mauritania, West Africa," has been formally accepted for publication in PLOS Neglected Tropical Diseases.

Best regards,

Shaden Kamhawi

co-Editor-in-Chief

Paul Brindley

co-Editor-in-Chief
